# Modelling the Dynamics of Multiagent $Q$-Learning in Repeated Symmetric Games: a Mean Field Theoretic Approach

**Shuyue Hu, Chin-Wing Leung, Ho-fung Leung**
The Chinese University of Hong Kong, Hong Kong, China
{syhu,cwleung,lhf}@cse.cuhk.edu.hk

## Abstract

Modelling the dynamics of multi-agent learning has long been an important research topic, but all of the previous works focus on 2-agent settings and mostly use evolutionary game theoretic approaches. In this paper, we study an $n$-agent setting with $n$ tends to infinity, such that agents learn their policies concurrently over repeated symmetric bimatrix games with some other agents. Using the *mean field theory*, we approximate the effects of other agents on a single agent by an averaged effect. A *Fokker-Planck* equation that describes the evolution of the probability distribution of $Q$-values in the agent population is derived. To the best of our knowledge, this is the first time to show the $Q$-learning dynamics under an $n$-agent setting can be described by a system of only three equations. We validate our model through comparisons with agent-based simulations on typical symmetric bimatrix games and different initial settings of $Q$-values.

## 1 Introduction

A multi-agent system concerns a set of autonomous agents interacting in a shared environment. Learning in multi-agent systems has recently attracted much attention [3, 13, 15], since multi-agent systems find application in a wide variety of domains, such as traffic control [1], energy management [20], robotic coordination [19], and distributed sensing [16]. While single-agent reinforcement learning has acquired a strong theoretical foundation [26], there is a lack of a thorough understanding of reinforcement learning under multi-agent settings [2]. Shoham [24] calls for more grounded research in this area rather than designing arbitrary learning strategies that result in convergence to a certain solution concept. Bloembergen et al. [2] point out that the modelling of multi-agent learning dynamics may facilitate parameter tuning, systematic comparison of different learning algorithms, and shedding light into the design of new learning algorithms.

Tuyls et al. [27, 29] model the dynamics of $Q$-learning with Boltzmann exploration in repeated 2-player bimatrix games using a evolutionary game theoretic approach. They derive a differential equation for each of the row and column player, and show that the learning process of each player can be understood as the replicator dynamics of a strategy change in an infinitely large agent population. Extensions have been made to study the dynamics of other learning algorithms, such as FAQ-learning [10], lenient FAQ-learning [18], gradient ascent [9] and regret minimization [12], in a similar manner. Gomes and Kowalczyk [22] construct a continuous time model for $Q$-learning, but focus on how another exploration strategy, $\epsilon$-greedy, affects the expected behaviours of agents. Wunder et al. [32] use dynamical system methods to study an idealization of $Q$-learning with $\epsilon$-greedy in repeated 2-player general-sum games. They show that the use of this learning method in certain subclasses of general-sum games induces chaotic behaviour for some initial conditions.

In general, all of the aforementioned works focus on the dynamics of reinforcement learning under 2-agent settings. Many real-life multi-agent systems, however, involve a much greater number of agents by nature. In this paper, we consider an $n$-agent setting with $n$ tends to infinity, such that, concurrently, agents learn their policies over repeated 2-player symmetric bimatrix games with some other agents in the population. The opponents that an agent interacts with will change from time to time. Thus, instead of learning against some fixed opponents, an agent learns to play with a wide range of socially changeable opponents. We note that this scenario, which has not been considered in the literature before, is a typical setting in norm emergence research [23].

One major difficulty of modelling multi-agent learning dynamics is to cope with non-stationarity, i.e., the fact that the interactions of agents leads to a highly dynamic shared environment [2, 28, 22]. One can expect that this non-stationarity will drastically increase as the total number of agents increases. This makes directly applying previous models in aforementioned works to $n$-agent settings inappropriate, because, in principle, the number of equations required to model the entire population dynamics is proportional to the number of agents in the population. As $n$ tends to infinity, analyzing or solving this system of equations becomes practically infeasible. We find the *mean field theory* [31] in statistical mechanics sheds light on this kind of problems. According to this theory, all of the effects of neighboring particles impose on a single particle can be approximated by an averaged effect—*mean field*—on that particle. This consequently reduces the degrees of freedom of the problem, and may make the problem analytically solvable.

Here, we assume agents use $Q$-learning with Boltzmann exploration. Using the mean field theory, we approximate the effects of other agents on a single agent with an averaged effect, such that one can conceive each agent in effect learns its policy over repeated interactions with a fictitious agent using the mean policy of the population. The $Q$-learning processes of individual agents will change the environment shared by all the agents. To capture this effect, we derive a *Fokker-Planck* equation that describes how the distribution of $Q$-values of the entire population evolves as time goes forward. We show under the $n$-agent setting we consider, the population dynamics can be modelled by a system of only three equations. For validation, we compare the behaviours obtained by our mean field theoretic model with the behaviours found in agent-based simulations. The comparison indicates our model well describes the qualitatively different patterns of evolution resulting from different types of symmetric bimatrix games and different initial settings of $Q$-values.

It is interesting to note that there is another line of research [17, 25, 33] on reinforcement learning in mean-field games [8, 14]. In this line of research, novel learning algorithms that converge to certain solution concepts (e.g., Nash equilibria) in mean-field games are proposed, however, the actual process of convergence is not formally described. This paper, to the best of our knowledge, is the first time to formally show the reinforcement learning dynamics in an infinitely large agent population. In particular, the Fokker-Planck equation describing the evolution of the probability distribution of $Q$-values in an agent population has not been reported elsewhere.

## 2 Preliminaries

In this paper, we focus on an infinitely large $Q$-learning agent population, in which each agent learns its policy concurrently over repeated symmetric bimatrix games with some other agents. Here we present the $n$-agent learning framework we consider in Section 2.1. The necessary backgrounds on symmetric bimatrix games and $Q$-learning are provided in Sections 2.2 and 2.3 respectively.

### 2.1 An $n$-Agent Concurrent Learning Framework

Consider a large population $\mathcal{N} = \{1, \ldots, n\}$ of $n$ agents, where $n$ tends to infinity. Each agent has the same set $\mathcal{A} = \{a_1, \ldots, a_k\}$ of $k$ available actions for a symmetric bimatrix game $G$. The learning framework of these $n$ agents is presented in Algorithm 1. Specifically, at each time step, an individual agent first independently selects an action to use according to its own policy (lines 3-5). Then, each agent plays the game $G$ with each of the $m$ opponents that are randomly selected from the population (lines 6-12). Note that the $m$ opponents with whom an agent play games may change for different time steps. A larger value of $m$ suggests agents can learn their policies from the interactions with a wider range of other agents.[1] For normalization, we assume the received immediate payoff for an

individual agent is averaged over all of the $m$ games it plays at each time step. At the end of each time step, each agent learns its policy independently and concurrently, so as to maximize its own future payoff (lines 13-16).

---

**Algorithm 1** An $n$-Agent Concurrent Learning Framework

---

**Require:** a set $\mathcal{N}$ of agents, a set $\mathcal{A}$ of available actions, a symmetric game $G$, the number $m$ of opponents per time step, the maximum time step $T$
 1: **while** $t < T$ **do**
 2:     $t \leftarrow t + 1$
 3:     **for** each agent $i \in \mathcal{N}$ **do**
 4:         Agent $i$ select an action $a \in \mathcal{A}$ according to its policy
 5:     **end for**
 6:     **for** each agent $i \in \mathcal{N}$ **do**
 7:         **while** $\theta_i < m$ **do**                    ▷ $\theta_i$ is the number of games that agent $i$ has played
 8:             Randomly select agent $j$ from the population $\mathcal{N}$ if $\theta_j < m$
 9:             Agents $i$ and $j$ play the game $G$ using their selected actions respectively
10:             $\theta_i \leftarrow \theta_i + 1, \theta_j \leftarrow \theta_j + 1$
11:         **end while**
12:     **end for**
13:     **for** each agent $i \in \mathcal{N}$ **do**
14:         Receive an immediate payoff, and update its policy using a learning method
15:         $\theta_i \leftarrow 0$
16:     **end for**
17: **end while**

---

## 2.2 Symmetric Bimatrix Games

Bimatrix games are typical mathematical modellings of strategic interactions between rational decision-makers (agents). Conventionally, in such a game, there are two players: the row player and the column player. The players play an action at the same time, and receive a payoff immediately. A bimatrix game is symmetric if both the players have the same set of available actions, and the resulting payoff of each player depends not on the role of the player, but only on their joint actions [4]. For reasons of exposition, we focus on an action set size of 2. In Table 1, we present a general form of 2-player-2-action symmetric bimatrix games. The first number of each entry is the payoff of the row player and the second number is the payoff of the column player. Clearly, the payoff matrix of the row player is the transpose of the payoff matrix of the column player.

| | Action $a_1$ | Action $a_2$ |
|---|---|---|
| Action $a_1$ | $\alpha, \alpha$ | $\beta, \gamma$ |
| Action $a_2$ | $\gamma, \beta$ | $\delta, \delta$ |

Table 1: A general form of 2-player-2-action symmetric bimatrix games.

## 2.3 $Q$-Learning for Bimatrix Games

$Q$-learning [30] is one of the most important algorithms in reinforcement learning research, and is the basis of a number of multi-agent reinforcement learning algorithms [3, 5, 7]. Given that there is a set $\mathcal{S}$ of states and a set $\mathcal{A}$ of available actions, such that an agent may transit to a new state $s' \in \mathcal{S}$ as a result of using an action $a \in \mathcal{A}$ under the current state $s \in \mathcal{S}$. $Q$-learning maintains a $Q$-value for each state-action pair $(s, a)$ to estimate the cumulative payoff over the successive time steps after performing action $a$ at state $s$. Consider an arbitrary agent $i$ in the agent population $\mathcal{N}$. Suppose that at time $t$, it plays the $j$th action $a_j$ under state $s$, and receives an immediate payoff $r_t^i(s, a_j)$ accordingly. This agent will update its $Q$-value $Q_{t+1}^i(s, a_j)$ for the state-action pair $(s, a_j)$ as follows:

$$Q_{t+1}^i(s, a_j) = (1 - \eta)Q_t^i(s, a_j) + \eta[r_t^i(s, a_j) + \gamma \max_{\forall a' \in A} Q_t^i(s', a')], \qquad (1)$$

where $\eta$ is the learning rate, $\gamma$ is the discounting factor, and $s'$ is the resulting state of using action $a$ under state $s$, so that the term $\gamma \max_{a' \in A} Q_t^i(s', a')$ estimates the optimal discounted future payoff

after the state transition. For any bimatrix game, there is only one episode in the entire course (or round) of the game: at a given time step $t$, the players each takes one action simultaneously, and receives an immediate payoff; then, the game ends. At the next time step $t + 1$, agents play another round of the game. This means that from time $t$ to $t + 1$, there is no state transition for an agent in bimatrix games, and the resulting state $s'$ does not exist at all. Therefore, for bimatrix games, it is a common practice to maintain a vector of $Q$-values for each action, i.e., $\mathbf{Q}_t^i = [Q_t^i(a_1) \ \ldots \ Q_t^i(a_k)]^\mathsf{T}$, and remove the term $\gamma \max_{a' \in A} Q_t^i(s', a')$ from the $Q$-value update function [11, 22, 32]:

$$Q_{t+1}^i(a_j) = (1 - \eta)Q_t^i(a_j) + \eta r_t^i(a_j). \tag{2}$$

We consider that each agent adopts a mixed-strategy policy, such that its $Q$-values are interpreted as Boltzmann probabilities for action selection. Let $\mathbf{x}_t^i = [x_t^i(a_1) \ \ldots \ x_t^i(a_k)]^\mathsf{T}$ represent the mixed-strategy policy of agent $i$ at time $t$, in which each component $x_t^i(a_j), \forall a_j \in \mathcal{A}$ is its probability of playing action $a_j$ at time $t$. The value of $x_t^i(a_j)$ is given as follows:

$$x_t^i(a_j) = \frac{e^{\tau Q_t^i(a_j)}}{\sum_{\forall a \in \mathcal{A}} e^{\tau Q_t^i(a)}}, \tag{3}$$

where $\tau \in [0, \infty)$ is the Boltzmann exploration temperature. A larger value of $\tau$ indicates the fewer exploration for individual agents. When $\tau$ is 0, the probability of taking each action is uniform, which means that agents take actions randomly. When $\tau \to \infty$, agents take the action with the highest $Q$-value in probability 1.

## 3   A Mean Field Theoretic Model

In this section, we model the $Q$-learning dynamics under the $n$-agent setting presented in the last section. In Section 3.1, taking the view of an individual agent, we model the dynamics of its $Q$-values with mean field approximation, such that, fictitiously, an agent updates its $Q$-values in response to the mean policy of the population. In Section 3.2, taking an bird eye's view, we model how the probability distribution of $Q$-values in the population evolves as time goes forward, and show the population dynamics can be characterized by a system of three equations.

### 3.1   Dynamics of $Q$-values for Individual Agents

Consider an arbitrary agent $i$ in the population $\mathcal{N}$. By Equation 2, we can derive the difference equation of its $Q$-values in terms of expected change. For any action $a_j$, at time $t$, the expected change of the corresponding $Q$-value is given as follows:

$$\begin{aligned} \mathbb{E}[Q_{t+1}^i(a_j) - Q_t^i(a_j)] &= x_t^i(a_j)[Q_{t+1}^i(a_j) - Q_t^i(a_j)] + [1 - x_t^i(a_j)] \times 0 \\ &= \eta x_t^i(a_j)[\mathbb{E}[r_t^i(a_j)] - Q_t^i(a_j)]. \end{aligned} \tag{4}$$

On the right hand side of the first line, the first term represents the change in the $Q$-value if action $a_j$ is used at time $t$, and the second term indicates that there should be no change in the $Q$-value if action $a_j$ is not used at time $t$. In the continuous time limit, this difference equation corresponds to the following differential equation:

$$\begin{aligned} \mathbb{E}\left[\frac{dQ_t^i(a_j)}{dt}\right] &= \eta x_t^i(a_j)[\mathbb{E}[r_t^i(a_j)] - Q_t^i(a_j)] \\ &= \eta \frac{e^{\tau Q_t^i(a_j)}}{\sum_{\forall a \in \mathcal{A}} e^{\tau Q_t^i(a)}}[\mathbb{E}[r_t^i(a_j)] - Q_t^i(a_j)]. \end{aligned} \tag{5}$$

This differential equation governs the dynamics of the expected change in $Q$-values for individual agents. By this equation, at a certain time step $t$, how fast an agent increases or decreases its $Q$-value for a particular action is susceptible to the learning rate $\eta$, Boltzmann exploration temperature $\tau$, the current $Q$-values and the received payoff at this time step.

Remember that at each time step, an agent play games with $m$ other agents that are randomly selected from the population. Let us first focus on one particular round of the game $G$ and assume the opponent in this round to be agent $z$. We denote the payoff matrix of the row player in game $G$ by $\mathbf{U}$. For agent $i$, the expected payoff of taking action $a_j$ against agent $z$ using the policy $\mathbf{x}_t^z$ is determined as:

$$\mathbb{E}[r_t^i(a_j, \mathbf{x}_t^z)] = \mathbf{e}_j^\mathsf{T} \mathbf{U} \mathbf{x}_t^z, \tag{6}$$

where $\mathbf{e}_j$ is the unit vector, in which the $j$th component equals 1 and the other components equal 0. Let $\bar{\mathbf{x}}_t = [x_t(a_1) \ \ldots \ x_t(a_k)]^\intercal$ be the mean policy of the population $\mathcal{N}$ at time $t$, such that $\bar{\mathbf{x}}_t = \frac{1}{n} \sum_{\forall i \in \mathcal{N}} \mathbf{x}_t^i$, where $n$ is the total number of agents. The policy $\mathbf{x}_t^z$ of agent $z$ can be represented by a deviation $\Delta \mathbf{x}_t^z$ from the mean policy $\bar{\mathbf{x}}_t$, such that $\mathbf{x}_t^z = \bar{\mathbf{x}}_t + \Delta \mathbf{x}_t^z$. With the first-order Taylor series expansion, the expected payoff $\mathbb{E}[r_t^i(a_j, \mathbf{x}_t^z)]$ is approximated as:

$$\begin{aligned} \mathbb{E}[r_t^i(a_j, \mathbf{x}_t^z)] &= \mathbb{E}[r_t^i(a_j, \bar{\mathbf{x}}_t + \Delta \mathbf{x}_t^z)] \\ &\approx \mathbb{E}[r_t^i(a_j, \bar{\mathbf{x}}_t)] + \mathbb{E}[\nabla r_t^i(a_j, \bar{\mathbf{x}}_t)^\intercal \Delta \mathbf{x}_t^z]. \end{aligned} \tag{7}$$

Let $\mathcal{M}_t^i \subset \mathcal{N}$ be the set of $m$ opponents that agent $i$ plays games with at time $t$. For agent $i$, its expected received payoff of taking action $a_j$ at time $t$, that is, $\mathbb{E}[r_t^i(a_j)]$, which is averaged over the $m$ rounds of games it plays with $m$ opponents, is approximated as:

$$\begin{aligned} \mathbb{E}[r_t^i(a_j)] &= \frac{1}{m} \sum_{z \in \mathcal{M}_t^i} \mathbb{E}[r_t^i(a_j, \mathbf{x}_t^z)] \\ &\approx \frac{1}{m} \sum_{z \in \mathcal{M}_t^i} [\mathbb{E}[r_t^i(a_j, \bar{\mathbf{x}}_t)] + \mathbb{E}[\nabla r_t^i(a_j, \bar{\mathbf{x}}_t)^\intercal \Delta \mathbf{x}_t^z]] \\ &= \mathbb{E}[r_t^i(a_j, \bar{\mathbf{x}}_t)] + \nabla r_t^i(a_j, \bar{\mathbf{x}}_t)^\intercal \mathbb{E}[\frac{1}{m} \sum_{z \in \mathcal{M}_t^i} \Delta \mathbf{x}_t^z] \\ &\approx \mathbb{E}[r_t^i(a_j, \bar{\mathbf{x}}_t)]. \end{aligned} \tag{8}$$

As the value of $m$ increases, the term $\mathbb{E}[\frac{1}{m} \sum_{z \in \mathcal{M}_t^i} \Delta \mathbf{x}_t^z]$ will become closer to 0, and hence the approximation will become more accurate.[2] By this approximation, for an individual agent, its received payoff of playing with its opponents is approximately the payoff of playing against the mean policy $\bar{x}_t$ averaged over all of the agents in the population. That is to say, although different agents actually interact with different opponents, intuitively, one can conceive different agents face one same fictitious agent that uses the mean policy.

Substituting the term $\mathbb{E}[r_t^i(a_j)]$ with the approximation shown in Equation 8, Equation 5—the equation that fundamentally governs the dynamics of the expected change in $Q$-values for individual agents—is rewritten as follows:

$$\mathbb{E}[\frac{dQ_t^i(a_j)}{dt}] = \eta \frac{e^{\tau Q_t^i(a_j)}}{\sum_{\forall a \in \mathcal{A}} e^{\tau Q_t^i(a)}} [\mathbb{E}[r_t^i(a_j, \bar{\mathbf{x}}_t)] - Q_t^i(a_j)]. \tag{9}$$

On the right hand side, the learning rate $\eta$ and Boltzmann exploration temperature $\tau$ are a priori given and the same for the entire agent population. Moreover, for symmetric bimatrix games, the expected payoff of using action $a_j$ against the mean policy $\bar{\mathbf{x}}_t$ is independent of the roles of individual agents. Therefore, at time $t$, for any agent $i$, how fast it changes its $Q$-values should be attributed to its current $Q$-values $\mathbf{Q}_t^i$ and the mean policy $\bar{\mathbf{x}}_t$ of the whole population. Dropping the agent index, for any individual agent in the population, Equation 9 can be expressed as a function $v_j$ of its current $Q$-values and the mean policy:

$$v_j(\mathbf{Q}_t, \bar{\mathbf{x}}_t) \triangleq \mathbb{E}[\frac{dQ_t(a_j)}{dt}] = \eta \frac{e^{\tau Q_t(a_j)}}{\sum_{\forall a \in \mathcal{A}} e^{\tau Q_t(a)}} [\mathbb{E}[r_t(a_j, \bar{\mathbf{x}}_t)] - Q_t(a_j)]. \tag{10}$$

Note that the mean policy $\bar{\mathbf{x}}_t$ is indeed given by the $Q$-values of all agents in the population, i.e., $\forall a_j \in A, \bar{x}_t(a_j) = \frac{1}{n} \sum_{\forall i \in \mathcal{N}} x_t^i(a_j) = \frac{1}{n} \sum_{\forall i \in \mathcal{N}} (e^{\tau Q_t^i(a_j)} / \sum_{\forall a \in \mathcal{A}} e^{\tau Q_t^i(a)})$. Therefore, the expected change in $Q$-values for any individual agent is determined by the joint $Q$-values of all the agents, which include the $Q$-values of itself. This suggests that in long term, the trajectories of $Q$-values for individual agents are uniquely determined by their joint initial $Q$-values.

## 3.2 Evolution of the Distribution of $Q$-values in a Population

Consider a $Q$-value space $\mathbb{R}^k$ with $k$ axes $Y_1, \ldots, Y_k$, where $k$ is the number of available actions. At time $t$, each agent $i$ occupies a point $\mathbf{Q}_t = \mathbf{Q}_t^i$ in this space according to its current $Q$-values $\mathbf{Q}_t^i$.

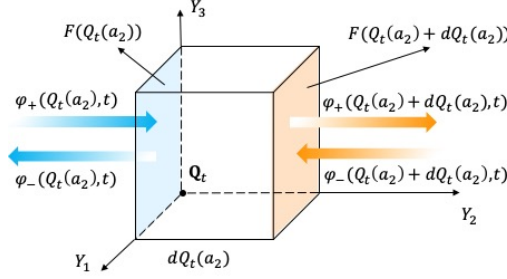

Figure 1: A 3-dimensional illustration of the entry and departure of individual agents through facets causing the change in the number of agents in the box $B$.

Let $p(\mathbf{Q}_t, t)$ be the function of agent density in the space at time $t$, such that the density $p(\mathbf{Q}_t, t)$ at any point $\mathbf{Q}_t$ is the proportion of agents in the population having their $Q$-values equal to $\mathbf{Q}_t$ and hence occupying the point $\mathbf{Q}_t$ in the space at time $t$. Intuitively, $p(\mathbf{Q}_t, t)$ can also be considered as the probability distribution of $Q$-values in the agent population. Note that agents will update their $Q$-values during interactions. As a result, as time $t$ moves forward, agents will change their positions in the space, which will lead to the change in the density function $p(\mathbf{Q}_t, t)$. In what follows, we shall derive the differential equation that describes the time evolution of $p(\mathbf{Q}_t, t)$.

Let us focus on an arbitrary point $\mathbf{Q}_t$ in this space, and an infinitesimal box (or hyperrectangle) $B$ around this point, such that $B \triangleq \{\mathbf{q}_t : Q_t(a_j) \leqslant q_t(a_j) \leqslant Q_t(a_j) + dQ_t(a_j), \forall a_j \in \mathcal{A}\}$. Basically, the number of agents in this box at time $t$ is $np(\mathbf{Q}_t, t)dV$, where $dV = \Pi_{\forall a_j \in \mathcal{A}} dQ_t(a_j)$ is the volume of the box. Given that there is no birth or death of individual agents over time, there is only one cause for the change in the density $p(\mathbf{Q}_t, t)$ of agents in that box—some agents enter or leave the box through its surface. Note that there are $2k$ facets for a $k$-dimensional box. Let $F(Q_t(a_j))$ denote a facet of this box, in which the $j$th component of each vector in this facet is set to $Q_t(a_j)$, such that the $Y_j$-axis is the normal of this facet. That is, $F(Q_t(a_j)) \triangleq \{\mathbf{q}_t : q_t(a_j) = Q_t(a_j), Q_t(a_i) \leqslant q_t(a_i) \leqslant Q_t(a_i) + dQ_t(a_i), \forall i \in \{1, \dots, k\} \backslash \{j\}\}$. We define $\psi_+(Q_t(a_j), t)$ and $\psi_-(Q_t(a_j), t)$, respectively, to be the number of agents that travel through the facet $F(Q_t(a_j))$ in the positive and negative direction of the $Y_j$-axis from time $t$ to $t + dt$. A graphical demonstration with the number of available actions $k = 3$ is shown in Figure 1. By the conservation law of the number of agents in the population, we shall have:

$$np(\mathbf{Q}_t, t + dt)dV - np(\mathbf{Q}_t, t)dV = \sum_{j=1}^{k} \psi_+(Q_t(a_j), t) + \psi_-(Q_t(a_j) + dQ_t(a_j), t) \tag{11}$$
$$- \psi_-(Q_t(a_j), t) - \psi_+(Q_t(a_j) + dQ_t(a_j), t).$$

This equation expresses that the number of agents entering (or leaving) the box should be the sum of the number of agents entering (or leaving) through every facets. The first and the second term on the left hand side represent the numbers of agents in this box at time $t + dt$ and at time $t$, respectively. Thus, the left hand side corresponds to the change in the number of agents in the box from time $t$ to $t + dt$. On the right hand side, since agents that travel through the facet $F(Q_t(a_j) + dQ_t(a_j))$ in the negative direction of the $Y_j$-axis will in effect enter the box $B$ (as shown in Figure 1), the first two terms are the number of agents entering the box $B$ along the $Y_j$-axis. Symmetrically, the last two terms are the number of agents leaving that box along the $Y_j$-axis. Hence, the right hand side corresponds to the sum of the net number of agents entering the box along every axes. Let $\psi(Q_t(a_j), t) \triangleq \psi_+(Q_t(a_j), t) - \psi_-(Q_t(a_j), t)$, which denotes the flow of agents travelling through the facet $\mathcal{F}(Q_t(a_j))$. Equation 11 can be rewritten as:

$$np(\mathbf{Q}_t, t + dt)dV - np(\mathbf{Q}_t, t)dV = \sum_{j=1}^{k} \psi(Q_t(a_j), t) - \psi(Q_t(a_j) + dQ_t(a_j), t). \tag{12}$$

We now derive the form of $\psi(Q_t(a_j), t)$. Remember that how fast an agent increases or decreases its $Q$-value, i.e., the velocity of this agent in the $Q$-value space, is given by the function $v_j(\mathbf{Q}_t, \bar{x}_t)$ shown in Equation 10. From time $t$ to $t + dt$, the displacement that an agent around the point $\mathbf{Q}_t$ travels should be approximately $v_j(\mathbf{Q}_t, \bar{\mathbf{x}}_t)dt$. That is, roughly speaking, agents that travel through the facet $F(Q_t(a_j))$ along the $Y_j$-axis from time $t$ to $t + dt$ should be located in the adjacent box $B' \triangleq \{\mathbf{q}_t : Q_t(a_j) - v_j(\mathbf{Q}_t, \bar{\mathbf{x}}_t)dt \leqslant q_t(a_j) \leqslant Q_t(a_j), Q_t(a_i) \leqslant q_t(a_i) \leqslant Q_t(a_i) + dQ_t(a_i), \forall i \in$

$\{1, \ldots, k\} \backslash \{j\}\}$. Therefore, the value of $\psi(Q_t(a_j), t)$ should be:

$$\psi(Q_t(a_j), t) = np(\mathbf{Q}_t, t)v_j(\mathbf{Q}_t, \bar{\mathbf{x}}_t)dtdS_j, \tag{13}$$

where $dS_j = \Pi_{\forall a_i \in \mathcal{A} \backslash \{a_j\}} dQ_t(a_i)$ is the area of the facet $F(Q_t(a_j))$, so that $v_j(\mathbf{Q}_t, \bar{\mathbf{x}}_t)dtdS_j$ is the volume of of the box $B'$. Substituting $\psi(Q_t(a_j), t)$ in Equation 12 with Equation 13, and dividing both sides by $dV dt$, we have:

$$\frac{np(\mathbf{Q}_t, t + dt) - np(\mathbf{Q}_t, t)}{dt} = \sum_{j=1}^{k} \frac{np(\mathbf{Q}_t, t)v_j(\mathbf{Q}_t, \bar{\mathbf{x}}_t)dS_j - np(\mathbf{Q}_t + d\mathbf{Q}_t, t)v_j(\mathbf{Q}_t + d\mathbf{Q}_t, \bar{\mathbf{x}}_t)dS_j}{dV}$$

$$= \sum_{j=1}^{k} \frac{1}{dQ_t(a_j)} [np(\mathbf{Q}_t, t)v_j(\mathbf{Q}_t, \bar{\mathbf{x}}_t) - np(\mathbf{Q}_t + d\mathbf{Q}_t, t)v_j(\mathbf{Q}_t + d\mathbf{Q}_t, \bar{\mathbf{x}}_t)].$$
$$\tag{14}$$

This equation in the continuous limit corresponds to:

$$\frac{\partial p(\mathbf{Q}_t, t)}{\partial t} = -\sum_{j=1}^{k} \frac{\partial}{\partial Q_t(a_j)} [p(\mathbf{Q}_t, t)v_j(\mathbf{Q}_t, \bar{\mathbf{x}}_t)] = -\nabla \cdot (p(\mathbf{Q}_t, t)\mathbf{v}(\mathbf{Q}_t, \bar{\mathbf{x}}_t)), \tag{15}$$

where $\nabla\cdot$ is the divergence operator, and $\mathbf{v}(\mathbf{Q}_t, \bar{\mathbf{x}}_t)$ is a vector field (or the *flux*) in which the $j$th component is $v_j(\mathbf{Q}_t, t)$. This equation is the *Fokker-Planck* equation [6, 21] with zero diffusion. By this equation, the change in the density of agents occupying a certain point $\mathbf{Q}_t$ in the space, which is also the probability density of agents having certain $Q$-values $\mathbf{Q}_t$ in the population, is jointly determined by the current density $p(\mathbf{Q}_t, t)$ and the velocity $\mathbf{v}(\mathbf{Q}_t, \bar{\mathbf{x}}_t)$. Note that the velocity in $Q$-values depends on the mean policy $\bar{\mathbf{x}}_t$. By the law of large numbers, each component $\bar{x}_t(a_j), \forall a_j \in \mathcal{A}$ of the mean policy $\bar{\mathbf{x}}_t$ should be close to the expectation, which is given by:

$$\bar{\mathbf{x}}_t = \int \cdots \int \frac{e^{\tau Q_t(a_j)}}{\sum_{\forall a \in \mathcal{A}} e^{\tau Q_t(a)}} p(\mathbf{Q}_t, t)dQ_t(a_1) \ldots dQ_t(a_k). \tag{16}$$

Therefore, the $Q$-learning dynamics of an infinitely large agent population can be modelling by the following system of three equations:

$$\begin{cases} \dfrac{\partial p(\mathbf{Q}_t, t)}{\partial t} = -\sum_{j=1}^{k} \dfrac{\partial}{\partial Q_t(a_j)} [p(\mathbf{Q}_t, t)v_j(\mathbf{Q}_t, \bar{\mathbf{x}}_t)], \\[2ex] v_j(\mathbf{Q}_t, \bar{\mathbf{x}}_t) = \eta \dfrac{e^{\tau Q(a_j)}}{\sum_{\forall a \in \mathcal{A}} e^{\tau Q_t(a)}} [\mathbb{E}[r_t(a_j, \bar{\mathbf{x}}_t)] - Q_t(a_j)], \\[2ex] \bar{\mathbf{x}}_t = \int \cdots \int \dfrac{e^{\tau Q_t(a_j)}}{\sum_{\forall a \in \mathcal{A}} e^{\tau Q_t(a)}} p(\mathbf{Q}_t, t)dQ_t(a_1) \ldots dQ_t(a_k). \end{cases} \tag{17}$$

This system of equation by nature involves a backward-forward structure. For an individual agent, at a certain time instant, it reasons backward and updates its $Q$-values towards a better estimation of the best response action facing the current expected policy. Collectively, the current updates of $Q$-values for individual agents may result in a future $Q$-value distribution that is different from the current one. This will in the other way round cause a change in the expected policy, which will make agents' current best responses to the expected policy invalid in the future. Therefore, under the $n$-agent setting we consider, the $Q$-learning agents are usually myopic.

## 4 Experimental Validation

In this section, we compare the behaviours obtained by our mean field theoretic model with the behaviours obtained from agent-based simulations. For the model, we employ finite difference methods to solve the system of equations shown in Equation 17. For the agent-based simulations, we set the number $n$ of agents to $1,000$, and consider two cases of the number $m$ of opponents per time step: $m = 0.05n$ and $m = n - 1$. To smooth out the randomness, we run 100 simulations for each setting. For comparison, the learning rate $\eta$ is set to $0.1$ and the exploration temperature $\tau$ is set to 2 in both the model and the simulations.

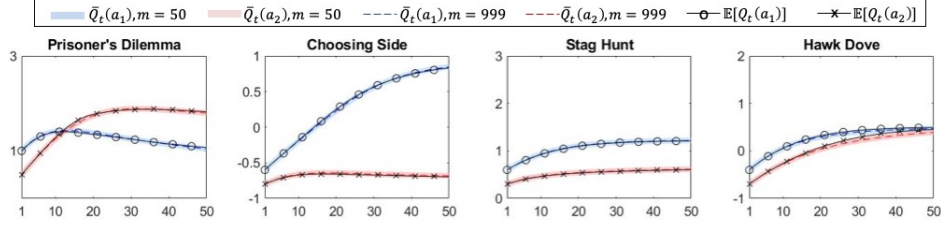

Figure 2: Evolution of the expected $Q$-values derived from our model and that of the mean $Q$-values in agent-based simulations.

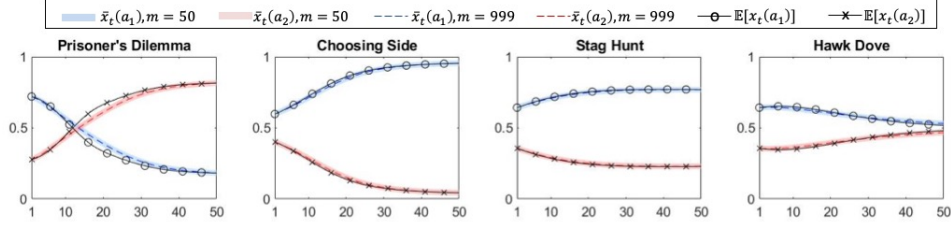

Figure 3: Evolution of the expected policy derived from our model and that of the mean policy in agent-based simulations.

To validate if our model can well reflect the diverse population dynamics caused by different game settings, we select four typical types of symmetric bimatrix games to experiment on, namely, prisoner's dilemma (PD), choosing side (CS), stag hunt (SH) and hawk dove (HD) games. The payoff bimatrices of these games are shown in Table 2. In PD game, the dominant strategy is for both players to play $D$, and hence $(D, D)$ is the unique Nash equilibrium. In CS game, there are two equally good symmetric Nash equilibria $(L, L)$ and $(R, R)$. In SH game, there are also two symmetric Nash equilibria, i.e., $(S, S)$ and $(H, H)$. However, while $(S, S)$ Pareto dominates $(H, H)$ and maximizes the social welfare, $(H, H)$ risk dominates $(S, S)$. In HD game, the two Nash equilibria $(D, H)$ and $(H, D)$ are asymmetric, such that it is unfair for the player taking $H$ in these two equilibria.

|   | $C$ | $D$ |
|---|-----|-----|
| $C$ | 3,3 | 0,5 |
| $D$ | 0,5 | 1,1 |

|   | $L$ | $R$ |
|---|-----|-----|
| $L$ | 1,1 | -1,-1 |
| $R$ | -1,-1 | 1,1 |

|   | $H$ | $S$ |
|---|-----|-----|
| $H$ | 1,1 | 2,0 |
| $S$ | 0,2 | 4,4 |

|   | $D$ | $H$ |
|---|-----|-----|
| $D$ | 1,1 | 0,2 |
| $H$ | 2,0 | -1,-1 |

(a) Prisoner's Dilemma, C: co-operate, D: defect    (b) Choosing Side, L: left, R: right    (c) Stag Hunt, S: stag, H: hare    (d) Hawk Dove, D: dove, H: hawk

Table 2: The typical symmetric bimatrix games that we experiment on.

Without loss of generality, for each game, we assume the initial $Q$-value of the first action and the second action follow Beta distributions $\text{Beta}(20, 80, r_{\min}, r_{\max})$ and $\text{Beta}(10, 90, r_{\min}, r_{\max})$, respectively. The first two parameters control the shape of the probability density function, and the latter two parameters prescribe the support to be $[r_{\min}, r_{\max}]$, where $r_{\min}$ is the minimum payoff of the game and $r_{\max}$ is the maximum payoff. Consequently, for every games, the initial expected $Q$-value of the first action is slightly higher than that of the second action.

In Figures 2 and 3, we compare the expected $Q$-values $\mathbb{E}[\mathbf{Q}_t]$ and the expected policy $\mathbb{E}[\mathbf{x}_t]$ obtained by our model with the counterparts $\bar{\mathbf{Q}}_t$ and $\bar{\mathbf{x}}_t$ that are averaged over all of the agents in the agent-based simulations. It is clear that our model well captures the qualitatively different patterns of evolution in agent populations playing different kinds of games. In particular, as shown in Figure 2, the dynamics of the expected $Q$-values generally overlap the dynamics of the mean $Q$-values, which suggest our model almost precisely describes how the $Q$-value distribution of the population evolves over time. Moreover, we note that in agent-based simulations, the agent behaviours with $m = 0.05n$ match those with $m = n - 1$. This implies that, strictly speaking, Equation 8 holds if $m, n \to \infty$, however, our mean field theoretic model should be practically valid if the values of $m$ and $n$ are sufficiently large.

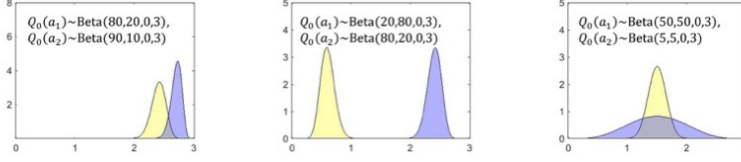

Figure 4: The probability density functions of different initial $Q$-value distribution that we experiment on. Yellow (light) color indicates the first action $H$ and purple (dark) color indicates the second action $S$.

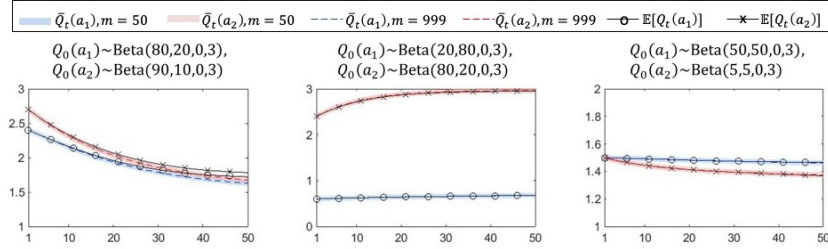

Figure 5: Evolution of the expected $Q$-values derived from our model and that of the mean $Q$-values in agent-based simulations.

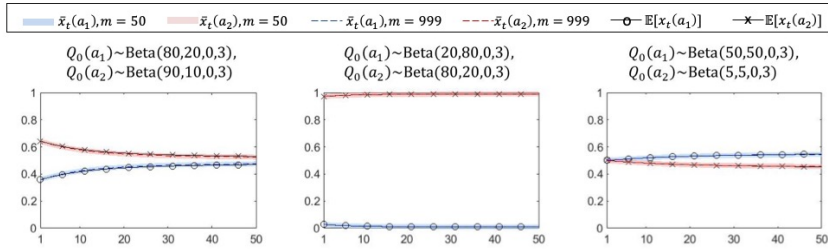

Figure 6: Evolution of the expected policy derived from our model and that of the mean policy in agent-based simulations.

We proceed to change the initial $Q$-value distribution for stag hunt games. Given the equilibrium $(S, S)$ is Pareto dominant but the other equilibrium $(H, H)$ is risk dominant, the population dynamics should be highly susceptible to the initial proportion of agents using each action. As shown in Figure 4, we consider three different cases of the initial $Q$-value distribution : 1) $Q_0(a_1) \sim \text{Beta}(80, 20, 0, 3)$ and $Q_0(a_2) \sim \text{Beta}(90, 10, 0, 3)$; 2) $Q_0(a_1) \sim \text{Beta}(80, 20, 0, 3)$ and $Q_0(a_2) \sim \text{Beta}(20, 80, 0, 3)$; and 3) $Q_0(a_1) \sim \text{Beta}(50, 50, 0, 3)$ and $Q_0(a_2) \sim \text{Beta}(5, 5, 0, 3)$. In Figures 5 and 6, we compare the expected $Q$-values and policy obtained by our model with the mean $Q$-values and policy in agent-based simulations. We can easily observe that the different settings of initial $Q$-value distribution results in diverse patterns of evolution in agent populations. Under each setting, the dynamics obtained by our model match those in agent-based simulations, which again validates our model well describes the population dynamics under different settings.

## 5   Conclusions and Future Work

In this paper, we model the dynamics of $Q$-learning in symmetric bimatrix games under an $n$-agent setting where $n \to \infty$. Using the mean field theory, we derive an equation that universally describes the dynamics of $Q$-values for any individual agent. We also derive a *Fokker-Planck* equation that describes the evolution of the distribution of $Q$-values in the agent population. We show the $Q$-learning dynamics under the $n$-agent setting can be described by a system of only three equations. The experiments on typical types of symmetric bimatrix games and different initial settings of $Q$-values validate that the expected agent behaviours obtained by our model well match the counterparts in agent-based simulations. As future work, we will extend our model to multiple-state games, asymmetric games, and multiple populations. Other learning algorithms will also be investigated.

## Footnotes

[1]When $m$ equals 1, our framework is in effect equivalent to the *social learning* [23], which is a commonly adopted framework in norm emergence research.

[2] The series in Equation 8 should be convergent, since the function $u(a_j, \mathbf{x}_t^z)$ is an analytic function. Given each element of the vector $\Delta \mathbf{x}_t^z$ is between 0 and 1, we consider the second order and the higher order terms negligible. When $m, n \to \infty$, Equation 8 holds.

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
