[Reviews · NeurIPS 2019]

Reviewer 1



The problem being tackled is that of describing the dynamics of Q-learning policies (through Q-values). This problem has been tackled before under a two player settings and this paper's main claim is its extension to n-player settings. The paper is well written and does seem to tackle a problem currently not reported elsewhere, however, I find the work lacking in two fronts: i) it is too derivative and incremental. Given the small contribution on the technical side, I would have liked a strong experimental section, but that is not the case and ii) experimental validation of the model is insufficient. The experiments are not showing anything insightful and (as I argue below) are not very useful. In order to improve this work, I suggest the authors to focus on developing good insights into the use of these methods for the design of new multiagent RL algorithms. In its current form I really do not see how the results presented here could be used in any way to design new algorithms. More so, given that there are already published works on multiagent RL in the mean field setting [Mguni et al. AAAI 2018, Mguni et al. AAMAS 2019], which by the way, these works are not even sited here, they should be used to validate this paper's models.

Reviewer 2



Let me start with a global comment. I enjoyed very much reading this paper. I found it well written (apart from typos, and some English sentences constructions that are a bit heavy) and interesting. It is related to a modern sub-field of reinforcement learning: multi-agent learning, that lacks theory w.r.t. to single-agent RL. The paper introduces a mean-field analysis of a large population of agents playing simple symmetric matrix games against each others, so that, as the population gets large, each player effectively plays against a single "mean" player. The theory is very well explained in spite of the conciseness of the paper, and the numerical experiments are very convincing. I recommend publication in NeurIPS. Let me give some more detailed comments (unfortunately the lines numbering are not present in the pdf; this is usually very convenient for pointing the mistakes, the authors will have to find where are the mistakes I'm referring to): _typos: page 2 (of AN individual agent may...; we introduce a probability...; of individual agentS under...). page 3 (Q-value WITH probability 1). page 4 (for AN individual agent; bar x_t should be bold; in fact irrelevant to the identity... -> in fact independent of the agents, but only depends on the current Q-values; such that given the settings of.. -> such that given...). page 5 (around the point q_t travel ALONG THE Z_i-AXIS; adjacent box B'={... for all j in {1,...k}/{i} } YOU MISS THE /{i}; of of the box). page 6 (recap -> recall; unfair for the player that plays H -> that plays D; table 2 (PD): left-down corner should be 5,0). page 8 (of AN individual Q-learner under A n-agent setting; dominates THE choice..) _please say in table 1 if row player's payoff is the first or second number in each entry of the payoff matrix _algo 1: it misses initializations: theta=0, t=0 (maybe others?) _my main question: From (1) to (2), because the game is stateless, the greedy term max Q (last term in (1)) disappears in the Q update equation. But I do not clearly see why, and this seems crucial for the analysis in the paper. This point, even if evident for the authors, should be clearly explained. _below (3): "where a=a_j.." is a useless sentence, as there is no a written above. _eq (4) is wrong as such: the second t+1 index on the left hand side should be t, and or 1) a should be replaced by a_j everywhere (the clearer solution to me, as the left hand side as it it written right now does not depend on j, while the right hand side does which is strange), or 2) you write explicitly close to (4) that a=a_j (the sentence below (3)). The same comment applies to the next equations, like (5) etc, where a should be a_j. Overall its a very good paper from which I leaned a lot. I recommend for acceptance after my comments have been taken into account, especially the authors should explain better how to go from (1) to (2). It would also have been nice to provide a code of the experiments. I have read the authors' rebuttal and taken it into account.

Reviewer 3



Originality: I am not an expert on RL systems and so it is somewhat difficult for me to judge this topic. I would largely defer to other reviewers here. However, I really enjoyed the exposition and found the line of research original and interesting. Quality: To me, it seemed that the quality of the research was high. I was able to follow the theoretical development and found it compelling. I strongly believe that I could sit down and reproduce the work without much difficulty. One place where some more exposition / details might be useful was in the experimental portion. Could the authors elaborate on exactly how the simulations were performed? A few more minor questions: 1) Is eq. 5 (especially converting from Q_{t+1} - Q_t) only valid in the \eta\to0 limit? 2) In eq. 8, is the series convergent? Would it be worth analyzing the \Delta x^2 correction to make sure it is subleading? Are there any conditions aside from n, m\to\infty that need to be satisfied here? 3) Eq 9 -> 10. It seems like in eq. 9 there are n-decoupled equations describing the per-agent Q-values. However, I don’t see how to get from there to a single agent independent equation. I.e. if two agents start with different Q values, surely their Q-values will follow different trajectories? Is this just saying that “without a loss of generality we can consider a single agent”? Which is a statement I agree with. Clarity: This paper was very clearly written. There are a few places where the language is not idiomatic, but I was very impressed as a non-expert by how easy the paper was to follow. One minor question, in eq. 4 & 5 should we be considering a_j since x_{j,t} is the probability given to action a_j? Significance: Naively, it seems to me that this work could be quite significant given increased interest in ever larger multi-agent games. Having said that, the action / state space considered here is small. It would be nice to include more examples including some that actually have state associated with them.

[Author Response · NeurIPS 2019]

**Response about the significance and originality.** Modelling the dynamics of multi-agent learning has long been an important research topic, but an $n$-agent setting where $n$ tends to infinity has not been considered. All of the previous works focus on 2-agent settings and mostly use evolutionary game theoretic approaches (see the recent survey [Bloembergen et al., JAIR'16]). Our mean field theoretic approach is of fundamental difference from the evolutionary game theoretic approaches. As we explain in Introduction, the use of evolutionary game theory is inappropriate for $n$-agent settings, because, in principle, the number of equations required to model the entire population dynamics is proportional to the number of agents in the population. As $n$ tends to infinity, analyzing or solving this system of equations becomes practically infeasible. In this paper, we show that by using mean field theory, only three equations are required to describe the dynamics of the whole population. A system of such small number of equations, as presented in Eq. 17, greatly reduces the problem complexity and makes the modelling tractable. Therefore, this paper introduces a new methodology to the modelling of learning dynamics in an infinitely large agent population, which is an emerging research topic given the growing interest in large-scale multi-agent systems.

The theoretical contributions of our work and the works [Mguni et al., AAAI'18; Mguni et al., AAMAS'19] mentioned by Reviewer 1 are very different. The works of Mguni et al. propose novel learning or incentive design methods, and prove that these methods will finally result in the convergence to efficient Nash equilibria in an infinitely large agent population. The actual process of convergence, however, is not formally described. This paper, to the best of our knowledge, is the first time to formally show the reinforcement learning dynamics, say, how the policies of individual agents gradually evolve over time, in an infinitely large agent population. In particular, the heart of this paper – a Fokker-Planck equation describing the evolution of the probability distribution of $Q$-values in an agent population – has not been reported elsewhere.

In this paper, we focus our attention on the population dynamics of an infinitely large agent population that use $Q$-learning. This is because $Q$-learning is one of the most important algorithms in reinforcement learning research and is the basis of a number of multi-agent reinforcement learning algorithms. Considering other learning algorithms will be an interesting and also plausible extension to our work.

We apologize that the above points should have been clearer. We shall highlight these points in the revised version.

**Response about the experiments.** The experimental study of this paper aims to illustrate and validate our mean field theoretic model. The games we select (prisoner's dilemma, stag hunt, hawk dove and choosing side) are typical matrix games that vary in the number, symmetry and efficiency of Nash equilibrium. This makes them good examples for illustration and validation, since they can be easily understood, but will lead to qualitatively different patterns of population dynamics. The nearly precise matching of the population dynamics derived from our model to those obtained from the agent-based simulations for each game type provides a clear and effective validation of our model. To further exhibit the strength of our model, we will find more complicated yet still understandable games to experiment on in the revised version. We will release our codes of the experiments if this paper get accepted.

**Response to the questions raised by Reviewer 2.** In each entry of Table 1, the first number is the payoff of the row player, while the second one is that of the column player. We shall include this specification in the revised version. The term $\gamma \max_{a' \in A} Q_t^{s', a'}(n_i)$ in Eq. 1 estimates the optimal discounted future payoff of player $n_i$ under state $s'$, after it plays action $a$ under the current state $s$ and consequently transits to the new state $s'$. For a matrix game, at a given time step $t$, agents play one round of the game. The row and the column players each takes one action simultaneously and receives an immediate payoff based on the joint actions. Then, the game ends. At the next time step $t + 1$, the agents play a new round of the game. In other words, from time $t$ to $t + 1$, there is no state transition for an agent. Since there is no state transition at all, there is no need to maintain the term $\gamma \max_{a' \in A} Q_t^{s', a'}(n_i)$. Hence, it is a common practice to remove this term from the $Q$-value update function for matrix games [Gomes and Kowalczyk, ICML'09; Wunder et al., ICML'10; Kianercy et al., Physical Review E'12]. We apologize for the lack of explanation in the current version, and shall provide a detailed one in the revised version. We appreciate a lot for pointing out our typos and giving valuable suggestions on the language!

**Response to the questions raised by Reviewer 3.** The agent-based simulations are conducted on 100 agent populations each consists of $1,000$ agents. Agents play games strictly following the interaction scheme presented in Algorithm 1, and use $Q$-learning to update their policies. In Eq. 4 and 5, $a$ should have been $a_j$. Eq. 5 holds for any valid value of $\eta$, which, by the definition of learning rate should be between 0 and 1. In Eq. 8, the series should be convergent, since the function $u(a, \mathbf{x}_t(n_j))$ is an analytic function. Given each element of the vector $\Delta \mathbf{x}_t(n_j)$ is between 0 and 1, we consider the second order and the higher order terms negligible. When $m, n \to \infty$, Eq. 8 holds. From Eq. 9, we can tell that the trajectory of each agent depends on its $Q$-values and is independent of who he/she is. Hence, we consider the trajectories of agents to be a function of $Q$-values in Eq. 10. We shall rewrite the left hand side of Eq. 10 to be $\mathbb{E}[\frac{dQ_t^{a_i}}{dt}](\mathbf{q}_t)$ for clarity. Given $n \to \infty$, the state of the population can be characterized by a distribution of $Q$-values in the population. Therefore, by deriving the Fokker-Planck equation that describes the time evolution of the $Q$-value distribution, we show in Eq.17 that only three equations are required to describe the entire population dynamics.

[Meta-Review · NeurIPS 2019]

This paper introduces a mean-field model of multiagent Q-learning in repeated symmetric games. The model assumes that at each time step each agent plays symmetric games with m other randomly chosen agents, and considers the limit of n, m to infinity. Under these settings the authors have derived the Fokker-Planck equation governing the time evolution of the distribution of the agents' Q-values. The review scores exhibited quite a large split. Two reviewers rated this paper well above the threshold, whereas Reviewer #1 rated it negatively. The main criticisms by Reviewer #1 are that this paper is derivative and incremental, and that the experiments are not sufficient. In my own view, however, the proposal of a mean-field model of multiagent Q-learning that allows description of its dynamics in terms of the Fokker-Planck equation is original enough as a theoretical contribution. I would therefore like to recommend acceptance of this paper, expecting that the authors will add citation to the already published work on multiagent RL in the mean-field setting, with discussion to put the contribution of this paper in view of them, as mentioned by Reviewer #1.